# Analysis of the Effect of the Use of Food Waste Disposers on Wastewater Treatment Plant and Greenhouse Gas Emission Characteristics

Dowan Kim 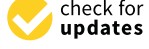 and Chaegun Phae *

Department of Environmental Engineering, Seoul National University of Science and Technology, Seoul 01811, Republic of Korea
* Correspondence: phae@seoultech.ac.kr; Tel.: +82-10-6218-6617

**Abstract:** The introduction of food waste disposers (FWDs) has been discussed in various countries, and in Korea, a method for utilizing FWDs has been considered. The results of the study show that the use of FWDs is more eco-friendly and economical than other forms of food waste (FW) disposal. However, there are also studies showing that FWDs are worse, for example, they aggravate water pollution and deteriorate the function of municipal wastewater treatment plants (WWTPs). Therefore, this study analyzed the concentration of pollutants of wastewater from FWD and the effect on the operation rate and GHG emission of WWTP when FW was introduced into the sewer by FWD using operation data for each WWTP. As a result of the analysis, when FWD was used, facilities exceeding the appropriate operation rate accounted for 86% of the total WWTP, and net-GHG emissions increased by 58%. Through this, FWD wastewater showed much higher contaminant concentrations than regular wastewater; thus, the introduction of FWD in the current situation will have a negative effect on maintaining the function of WWTP and reducing GHG. To introduce FWDs, improvement in WWTPs regarding pollutant load and discharge characteristics of FW and input of digestion systems through a separate FWD pipe, the introduction of high-efficiency energy facilities, and the recycling of wastewater sludge are necessary to reduce GHG emissions.

**Keywords:** food waste; food waste disposer; greenhouse gas; operation rate; municipal wastewater treatment plant



## 1. Introduction

Municipal wastewater treatment plants (WWTPs) are an integral part of urban systems. They require a significant amount of electricity and cost [1,2]. This energy and cost can be reduced by producing biogas. Methane ($CH_4$) and carbon dioxide ($CO_2$) are produced during anaerobic digestion (AD) [3,4]. Biogas can readily be used as a fuel to generate heat or electricity in WWTPs [5,6]. It should also be noted that wastewater and sludge treatment is associated with significant financial cost and energy for WWTPs. Sludge treatment accounts for 40–60% of total operating costs, and AD helps to reduce the amount of sludge produced [7]. Therefore, there is still a need for solutions that improve the recovery and production efficiency of electrical and thermal energy in WWTPs.

Significant amounts of food waste (FW) are generated worldwide, and typically exhibit the highest methane potential of any MSW component, resulting in significant fugitive greenhouse gas (GHG) emissions from landfills [8,9]. The landfill has been the most convenient and economical form of waste disposal in many countries around the world. However, increasing environmental regulations aimed at reducing the amount of biodegradable municipal waste and ensuring that FW goes to landfills have introduced alternative methods of managing FW.

To prevent FW from being disposed in landfills, many studies have been conducted worldwide to determine whether food waste disposers (FWDs) can be used [10,11]. They

are used with water while grinding to attain efficient and uniform grinding and discharge to the sewer; they are mainly used under the sink. As such, a vast majority of the FW produced at household level enters the wastewater stream and is treated in WWTPs. FWDs are used in many countries around the world, but their implementation as a waste management option has never been fully or appropriately considered.

WWTPs and FWD have a particularly important role to play in implementing circular economy goals. Food wastewater treated with FWDs flows to WWTP along the sewage pipe. Although some are caught on the screen, most are converted to sludge in the settling pond and thickener. This is fed into WWTPs' AD. FW can increase the biogas yield by increasing the concentration of organic matter in sludge with high organic matter and relatively low organic matter [12]. However, inconsistent and contradictory results have been reported for each study, and opinions on the introduction of FWD are divided around the world. this is due to reasons such as the properties of food, treatment process, and status of WWTP by country [13].

Looking at the positive studies on the introduction of FWD, it is argued that using FWD is a sustainable treatment method, because the amount of FW can be reduced by 42%, and wastewater can be treated in connection with the WWTP [14,15], which can reduce the cost of FW collection and increase biogas production [16]. A study on FWDs has also been conducted in Korea, and the method was found to be almost as sustainable as composting [17].

Looking at the negative studies on the introduction of FWDs, it is argued that introducing FW into the sewer increases the pollution load and may impair the soundness of wastewater treatment (including septic tanks), and is not an eco-friendly treatment method [18]. Household composting had the lowest impact on the environment in all impact categories, while FWDs consume less energy but have the highest eutrophication potential (the emission in WWTP), and water consumption is known to be the highest [19]. In addition, looking at the study on the effect of FWD on the wastewater system, it was reported that the BOD load increased by 17~62% and the SS increased by 1.9~7.1% [20]. In addition, other similar studies are concerned that COD, BOD, and T-N in FW may cause secondary loads [21].

The main cause of these conflicting research results can be seen as being influenced by the FW characteristics and analysis methods of the country concerned reviewing the introduction of FWD. The characteristics of FW and information on the wastewater system in Korea should be reflected and analyzed. However, similar studies conducted in Korea have been conducted on an experimental scale, and studies on the sewage system have not been theoretically reviewed, and only energy consumption has been analyzed for GHG reduction effects.

In Korea, FWDs are highly preferred, but their sale and use were banned in 1995, and manufacturing and importing were banned in 1999, considering the negative sewage impact. However, in recent years, the dissemination of the separated pipe has been expanded along with the claim to allow the use of FWDs. However, only discussions are being conducted on the pros and cons of the introduction, and objective data are needed to resolve this.

The purpose of this study was to assess the existing data on the potential impacts of the use of FWDs on the wastewater system and the environment. Therefore, to analyze, the effect of introducing FWD on the WWTP, an increase in inflow load on WWTPs with a capacity of 500 $m^3$/day or more, and the effects of GHG emission were analyzed.

## 2. Materials and Methods

### 2.1. Characteristics of Wastewater from Discharged Food Waste Disposer

In Korea, biochemical oxygen demand (BOD) and chemical oxygen demand (COD) are used as organic matter management indicators in effluents from WWTPs.

BOD is a method for measuring biodegradable organic substances among organic substances and has disadvantages in that it takes a lot of time for analysis and it is difficult

to accurately quantify the total amount of organic substances in a sample due to a low decomposition rate. In the case of COD, it can be a relatively accurate indicator of organic matter compared to BOD, but the difference in oxidation rate according to the properties of the sample is large and reproducibility is low, so it has been pointed out as a representative indicator [22]. On the other hand, total organic carbon (TOC) can accurately quantify the amount of organic matter in the water body by directly measuring the amount of carbon, reducing measurement errors and making it possible to measure non-degradable organic matter. Therefore, TOC in the water quality of food wastewater was also analyzed [23].

An experiment was conducted to understand the characteristics of the wastewater discharged from the FWD. In the experiment, a total of three types of wastewater were analyzed: wastewater discharged from washing/dehydration (WD), wastewater discharged from grinding (GR), and microbial liquid fermentation (MLF) types.

The WD type is a method used in general households, where the FW is placed in a sink, and natural dehydration is performed by gravity. Most of the FW will end up in the sieve. The GR type is a typical FWD method, but a sieve is installed at the end of the crushing part to recover 80% of the solids in FW. Theoretically, only 20% of the ground FW is discharged to the wastewater. The MLF type is where after grinding, the wastewater is transported to a microbial reactor, which is then decomposed by microorganisms and discharged into the sewage system. It is assumed that all solids are decomposed into water.

A wide range of geodemographic factors, including location, lifestyle, cooking habits, and socioeconomic conditions, can substantially affect the amount of MSW and thus the FW generated. The composition ratio of the sample was artificially produced using the composition ratio of "The national waste statistical survey 5th [24]", which can be seen in Table 1. A sample of 500 g was used. Water was injected at 3 L/40 s, wastewater was collected using the pipe, homogenized, and then collected and analyzed by 50 mL. However, since the MLF type requires a stabilization period, continuous experiments were conducted for 16 days. After measuring the pH of the collected sample, 10 mL of the sample was diluted 100 times and analyzed for TOC. The remaining filtrate was used to analyze TS and VS.

**Table 1.** Composition of sample used for analysis.

| Category | Processed Sample | |
| --- | --- | --- |
| | Weight (g) | Ratio (%) |
| Vegetables | Kimchi (68), lettuce (34), onion (34), radish kimchi (17) | 30.6 |
| Cereals | Rice (102) | 20.4 |
| Fruits | Tomato (17), orange peel (42), apple core (42) | 20.2 |
| Fish meat | Grilled pork belly (34), grilled fish (51) | 17 |
| Leachate | Soybean paste stew (34) | 6.8 |
| Etc. | Eggshell (17), banana peel (8) | 5 |
| Total | 500 g | 100 |

### 2.2. Analysis of the Impact on the Operation of WWTP

The use of an FWD increases the amount of wastewater treatment and pollution load in the WWTP, so it may be necessary to expand. The expansion was classified as subject to an expansion only when the utilization rate based on BOD load due to FWD exceeded 80%. Therefore, we reviewed the cost for the expansion and operation of WWTP (679 plants of 500 $m^3$/day or more) [25].

To analyze the effect of FWDs, the theoretical basic unit assuming the generation characteristics of FW and the amount of wastewater was applied. Theoretically, the amount of wastewater by using an FWD was applied at 15 L/day/household (5 L/1 time $\times$ 3 times a day in total). The BOD of FW is very different for each previous study [20] because it is desirable to use Korean research; BOD was applied as an average value (140 g/L) of 182.3 g/L [26], 107.9 g/L [27], 132.0 g/L [28], T-N was applied 4665 mg/L [29] (since there is only one study in Korea where wastewater T-N was analyzed).

The amount of FW generation was obtained in the national waste generation and treatment status by the Ministry of Environment [30]. The amount of FW inflow by WWTPs from FWDs was calculated by multiplying the sewage treatment population in the area covered by the WWTP by the amount of FW generated per household in the area. For example, the daily FW generation per household in Seoul is 0.62 kg, and if the number of households charged by WWTP A is 100, then 26 kg/day of FW inflow is applied.

*2.3. Analysis of GHG Emissions*

For the operation of FWDs, electricity is used and, energy is consumed (the energy consumed by FWDs may depend on the model, type, frequency, and duration of use). In addition, many studies have reported that the GHG caused by using FWD is negligible [19,31]. Therefore, in this study, GHG emissions generated during the FWD use stage were excluded from the analysis.

The effect of FWD on GHG emissions has been ignored in most studies. Increases in BOD, SS, and TKN can certainly increase GHG emissions in wastewater treatment processes, and increases in energy consumption and sludge transport, treatment, and disposal can also contribute to GHG emissions.

The wastewater treatment process includes two stages: mechanical (removal of suspended and suspended solids) and biological treatment (removal of organic contaminants, nitrogen, and phosphorus compounds). Mechanical treatment includes screening and removal of minerals and organic solids. Since the treatment process is different for each WWTP, the BOD material balance of the WWTP was used as a standard process in Korea and is shown in Figure 1. The BOD removal rate of the primary settling cell is 35%, and it is fed into the anaerobic digester in the form of concentrated primary sludge. Of the 65% of the BOD introduced into the bioreactor via the primary settling tank, 60% (total input BOD 39%) is oxidized (respiration of microorganisms), and 40% (26% of the total input BOD) is transferred to the secondary settling tank. Of the 26% of BOD, about 5% is discharged as effluent, about 10% is returned to the bioreactor, and the remainder is input to the Anaerobic digester in the form of concentrated surplus sludge from the secondary settling tank as surplus sludge. The BOD removal rate of the anaerobic digester is about 80%. Under this premise, if FW is treated using an FWD, it is brought in together with wastewater and passes through the bioreactor of a WWTP, 39% of the introduced BOD is decomposed, and only about 44.4% flows into the AD.

Assuming that organic matter is glucose ($C_6H_{12}O_6$), the amount of methane gas generated per 1 kg of BOD is 0.25 kg, and when converted to standard volume, it is 0.35 m$^3$ [32]. Therefore, the amount of methane gas generated was applied as 0.35 m$^3$ per inlet BOD. Emitted GHG from WWTP can be roughly divided into $CO_2$ generated using energy such as electricity and oil, and $CH_4$ and $N_2O$ emitted during water treatment. $CH_4$ is generated in anaerobic conditions during the water treatment process, and $N_2O$ is generated during the nitrification process, which is an aerobic condition, and the denitrification process, under anoxic conditions.

GHG can be generated in the water and sludge treatment processes. In the water treatment process, according to the IPCC guidelines (2006, Tier 1) [33], $CO_2$ emitted from wastewater is of biological origin and is excluded from the calculation of emissions, and $CH_4$, $CO_2$ was calculated according to the power required to remove $N_2O$ and BOD. IPCC Category 6D was applied as a methodology for calculating the amount of GHG emitted from biological treatment among GHG emitted from WWTP. To calculate, the parameters, the increased amount of wastewater and the pollution load were applied by adding the operation data of 679 WWTP, the amount of FW generated according to the population of each facility, and the amount treated by FWD.

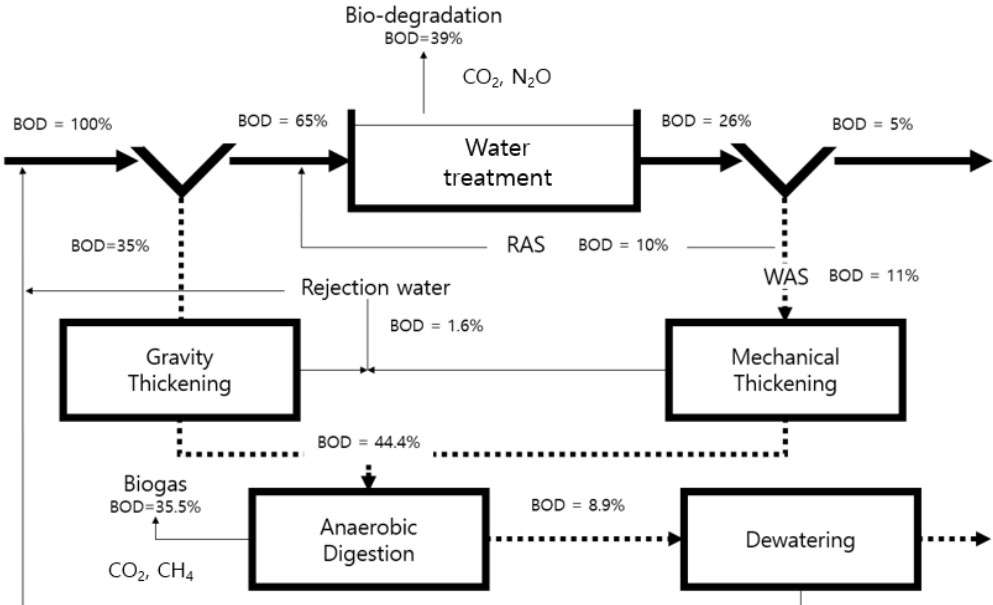

**Figure 1.** Material flow chart (BOD of wastewater) for wastewater treatment plants. RAS is returned activated sludge, WAS is wasted activated sludge.

The GHG calculation formula by IPCC is as follows (Table 2).

**Table 2.** GHG calculation formula for WWTP(Tier 1).

| Category | Contents |
|---|---|
| CH $_4$ Emissions | $(BOD_{in} \times Q_{in} - BOD_{Out} \times Q_{Out} - BOU_{sl} \times Q_{sl}) \times 10^{-6} \times EF - R$<br>Emissions from wastewater treatment (tCH4) |
| $BOD_{In}$ | Concentration of influent($BOD_5$) (mg-BOD/L) |
| $BOD_{out}$ | Concentration of effluent($BOD_5$) (mg-BOD/L) |
| $BOD_{sl}$ | Concentration of Sludge($BOD_5$) (mg-BOD/L) |
| $Q_{in}$ | Volume of influent (m$^3$) |
| $Q_{out}$ | Volume of effluent (m$^3$) |
| $Q_{sl}$ | Volume of sludge (m$^3$) |
| EF | Emission factor (kg CH$_4$/kg-BOD) 0.48 |
| R | Methane recovery (tCH$_4$) |
| N$_2$O Emissions | $(TN_{in} \times Q_{in} - TN_{Out} \times Q_{Out} - TN_{sl} \times Q_{sl}) \times 10^{-6} \times EF \times 1.571$<br>Emissions: emissions from wastewater treatment (tN2O) |
| $TN_{in}$ | Concentration of influent (mg-T-N/L) |
| $TN_{out}$ | Concentration of effluent (mg-T-N/L) |
| $TN_{sl}$ | Concentration of Sludge (mg-T-N/L) |
| EF | Emission factor (kg N$_2$O-N/kg-T-N) 1.571 |

The amount of FW was input for each WWTP, the increased rate of inflow concentration was applied, and the amount of wastewater generated, and FW input used in the data was prepared using the data used when reviewing the expansion WWTP that had been built as a database. For the analysis, it is assumed that an AD is installed in all domestic WWTPs and that all the generated sludge is incinerated.

## 3. Results

### 3.1. Effluent Characteristics Analysis of Food Waste Disposer

The TS of WD was 1.15%, 1.96% for the GR type, and 1.07~5.75% for the MLF type. In the microbial liquid fermentation type, a total of 61.9% of the input solids was liquefied and discharged (Table 3). When comparing the fact that the solids outflow rate of the MLF is 61.9% and the theoretically reviewed rate was 44.4%, some of the input solids seem

to be discharged without being decomposed. When compared with the reference values of wastewater, the dehydrated wastewater had higher TS and TOC than the wastewater (70.5~76.6 mg/L) [34]. On the other hand, the MLF type showed high concentrations of TOC other than TS. In the beginning, relatively clean wastewater was discharged, but as the experiment proceeded, the MLF type increased to 14,800 mg/L, and it was identified that the decomposition did not perform well due to the accumulation phenomenon inside the device(Figure 2). Considering that the concentration of TOC generally flowing into the WWTP is 1.7~293.7 [35], it is a very high concentration of wastewater. Increases in TSS and BOD/COD ratios due to the use of FWDs can lead to subsequent increases in sludge and biogas generation rates depending on the treatment processes followed in the WWTP.

**Table 3.** TS and TOC of wastewater.

| Category | Wash· Dehydration (1) | Type of FWD | |
| --- | --- | --- | --- |
| | | Grinding (2) | Microbial liquid fermentation (3) |
| **TS(%)** | 1.15 | 5.9 | 1.07~5.75 |
| **VS/TS** | 82 | 95 | 91~97.1 |
| **TOC (mg/L)** | 1680 | 5000 | 3724~14,800 |
| **pH** | 7.0 | 6.9 | 5.5~7.1 |

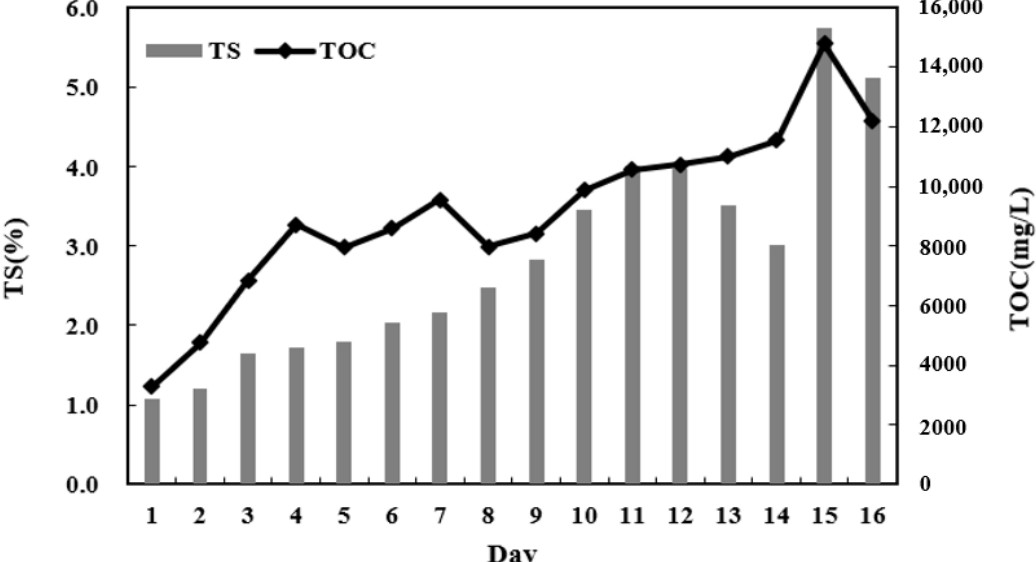

**Figure 2.** TS and TOC of wastewater from microbial liquid fermentation.

Comparing the pH, in the FWD, the MLF type of wastewater was lowered to pH 5.5. Potential increases in H production from a FWD can damage WWTP systems [36]. Sulphides are produced through anaerobic decomposition of organic matter in H wastewater. It is a known corrosive agent that can attack clay pipes, concrete, and metal surfaces, causing deterioration and subsequent leaks in the WWTP [37]. In addition, pH in the influent of WWTP has a great influence on microorganisms in the wastewater treatment process. Microorganisms in the wastewater treatment process are sensitive to pH changes, and in strong acid or alkali conditions, the activity of the microorganisms is reduced, and in severe cases, the cell walls and tissues of the microorganisms are destroyed, thereby destroying the wastewater treatment function [38].

In Figure 3a, the result of sedimentation of wastewater at 25 °C for 2 h is shown, and the degree of sedimentation of the FWD is more pronounced than that of washing and dewatering. Observation of the dry matter of the pulverized effluent for each method visually showed that fat, oil, and grease (FOG), and fine solids remained in the case of

WD, and a large amount of organic matter and oil remained in the FWD (Figure 3b). FOG contained in wastewater can solidify and stick to pipes, clogging the system [39]. Through the above experimental results, since wastewater contains pollutants, it seems unreasonable as wastewater.

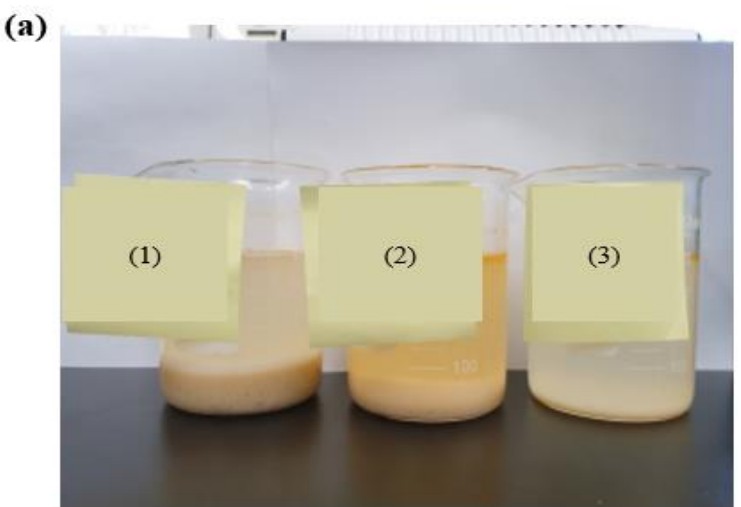

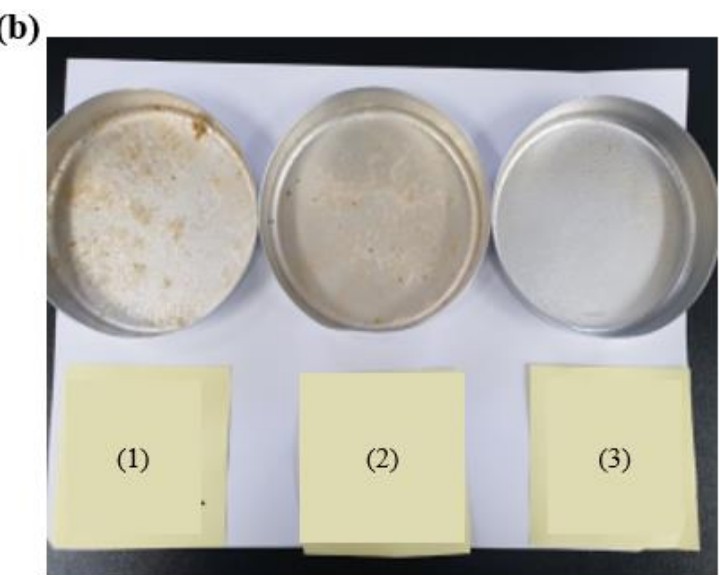

**Figure 3.** Sedimentation photograph of wastewater by FW treatment method (**a**) and dried photograph (**b**). (1) is WD, (2) is GR, and (3) is wastewater discharged from MLF type.

### 3.2. Analysis of the Impact on the Operation of WWTP

When a FWD was introduced based on the pollution load (BOD), the total inflow BOD load of domestic WWTP increased by 51.2%, and it was found to increase by 0.8~1.714% depending on the WWTP (Figure 4). Most of the facilities with a high rate of increase in the inflow BOD load were characterized as WWTP of less than 10,000 $m^3$. This is because these facilities are installed to provide only a minimal wastewater treatment function, so it is difficult to respond to an unplanned increase in load. However, WWTP with more than 100,000 $m^3$ often exceeds the design standard of BOD 180 ppm, so it may be difficult to handle FWDs even for a large WWTP.

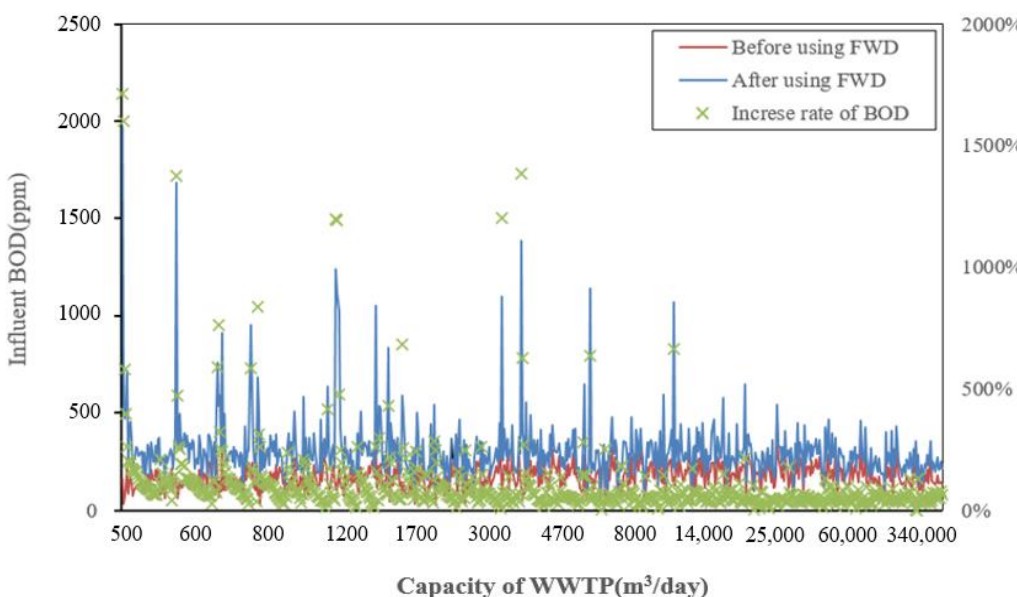

**Figure 4.** BOD concentration before and after the introduction of FWD by the capacity of WWTP.

As the concentration of incoming BOD increased due to FWDs, the average operation rate was to increase by 58%. Only 95 of the total 679 facilities had WWTP with an operation rate of less than 80% after using FWDs (Figure 5). This means that all facilities corresponding to the remaining 86% exceed the appropriate operation rate. If WWTP is operated with an overload, wastewater that has not been treated can be discharged into the river as it is, which can adversely affect the ecosystem. Therefore, it was found that 14% of facilities in Korea can introduce FWDs, and it is possible to introduce FWDs only in areas with lower population density than large cities. Additionally, average data were used in this analysis, but FW occurs intensively at specific times (morning, lunch, and dinner) and seasons, so the deviation will be very large. Due to this, the treatment efficiency is reduced due to the shock load of the bioreactor tank and lack of hydraulic residence time, and the scum and sludge overflow phenomenon may occur due to the lack of residence time of the settling tank [40].

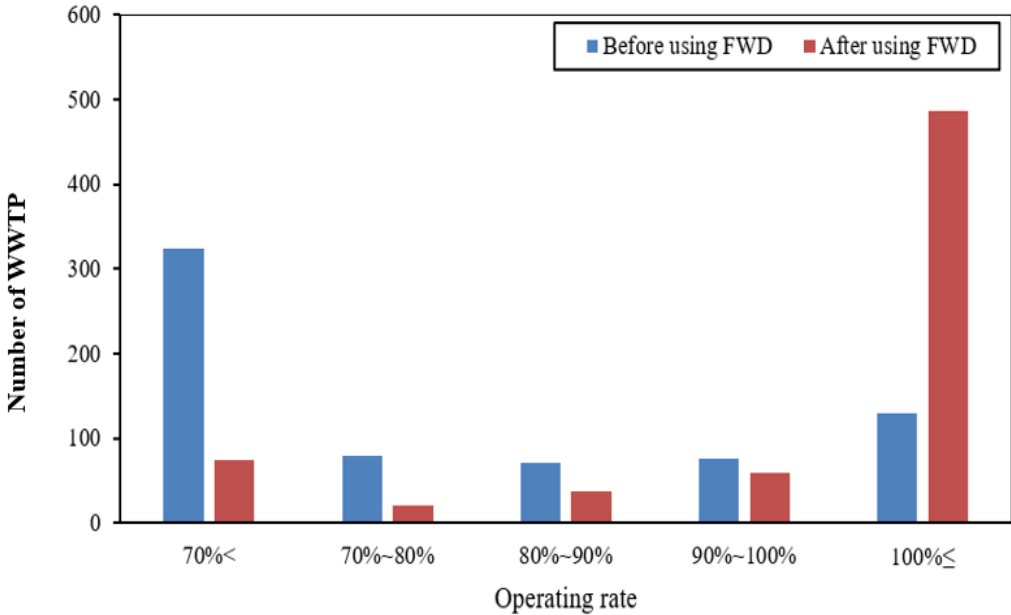

**Figure 5.** Changes in the WWTP utilization rate before and after the FWD introduction.

### 3.3. Analysis of GHG Emissions

Table 4 shows the results of the analysis of the GHG emission impact of the introduction of FWD. GHG emissions are −1854 $tCO_2eq$/day in the process of treating wastewater at a WWTP, and −3468 $tCO_2eq$/day when using FWDs. This is because $CO_2$ emitted from wastewater is an organic source and is excluded from the calculation of emissions and generates energy through anaerobic digestion. In the presence of the co-substrate, the removal efficiency of AD was improved. It improved from 64% (single digestion) to 69–70% (co-digestion) [41].

**Table 4.** GHG emissions from wastewater and wastewater + FWD treatment process.

| Category | | | Unit | Type | |
|---|---|---|---|---|---|
| | | | | Wastewater | Wastewater + FWD |
| Parameters | Influent | Q | $m^3$/day | 20,555,969 | 20,881,380 |
| | | FW | ton/day | 0 | 12,564 |
| | | Conc | mg-BOD/L | 167 | 251 |
| | | | mg-T-N/L | 41 | 43 |
| | Effluent | Q | $m^3$/day | 19,630,950 | 19,941,718 |
| | | Conc | mg-BOD/L | 4 | 4 |
| | | | mg-T-N/L | 10 | 10 |
| | Sludge | Q | ton/day | 12,471 | 19,263 |
| | | Conc | mg-BOD/L | 38,473 | 36,855 |
| | | | mg-T-N/L | 4306 | 3312 |
| EF-CH$_4$ by Wastewater | | | kgCH$_4$/kgBOD | 0.18452 | |
| EF-N$_2$O by Wastewater | | | kgN$_2$O/kgT-N | 0.00072 | |
| EF-GHG by Electric | | | kgCO$_2$eq/kW | 0.454 | |
| EF-CH$_4$ by Sludge | | | kgCH$_4$/ton | 0.0097 | |
| EF-N$_2$O by Sludge | | | kgN$_2$O/ton | 0.90 | |
| Wastewater treatment | | CH$_4$ GAS | $m^3$CH$_4$ | 960,656 | 1,465,568 |
| | | CH$_4$ emissions | tCH$_4$ | −157 | −239 |
| | | N$_2$O emissions | tN$_2$O | 4.7 | 5.0 |
| | | GHG emissions | tCO$_2$eq/day | −1854 | −3468 |
| Electric energy consumption | | Electric | KW/day | 8,718,673 | 13,800,886 |
| | | GHG emissions | tCO$_2$eq/day | 3958 | 6266 |
| Sludge disposal | | CH$_4$ emissions | tCH$_4$ | 0.12 | 0.19 |
| | | N$_2$O emissions | tN$_2$O | 11.2 | 17.3 |
| | | GHG emissions | tCO$_2$eq/day | 3482 | 5379 |
| Net-GHG emissions | | | tCO$_2$eq/day | 5586 | 8176 |
| Net-GHG emissions per ton of FW | | | kgCO$_2$eq/t-FW | - | 206.1 |

In terms of energy, additional power is required to remove contaminants flowing in from FW. The electricity required for this was calculated to be 8,718,673 KW/day when only wastewater is treated and 13,800,886 KW/day when FW is treated together. Converting this to the amount of GHG, it was found that 3958 $tCO_2eq$/day and 6266 $tCO_2eq$/day were generated, respectively.

The amount of wastewater sludge generated increased, and it was found that 12,471 tons/day of wastewater treatment alone and 19,263 tons/day of wastewater sludge were generated when FW was treated together. Assuming that it is incinerated, and applying the emission factor and global warming potential, it was found that the GHG emissions were 3482 $tCO_2eq$/day and 5379 $tCO_2eq$/day, respectively.

However, when only wastewater is treated, the net GHG emission is 5586 $tCO_2eq$/day, whereas when FW is treated together, the net GHG emission is 8176 $tCO_2eq$/day, which increases the GHG emission of the WWTP. Carbon neutrality is adversely affected when additional energy used for increased organic matter treatment is considered.

The net GHG emission per ton of FW is 206.1 $kgCO_2$/t-FW. This was higher than the research results of 125 $kgCO_2eq$/t-FW [42] and 121 $kgCO_2eq$/t-FW [17] when FW was treated through the conventional FWD. This is because the above two studies did not

even consider the GHG generated during the wastewater sludge treatment process, but only considered the energy used for simple FWD operation and wastewater treatment. In addition, there is also a low difference in the amount of pollution load by the used FW which is 76 g/kg-FW. However, considering that it is lower than 510 kgCO$_2$eq/t-FW (wastewater + FWD co-digestion) [43], if we add up the amount of GHG emitted into the atmosphere during the wastewater treatment process, it can be inferred that more GHGs are emitted.

However, these results do not mean that it is not rational to utilize FWDs. This is because GHG emissions can be reduced by actively installing energy-saving facilities or by composting and drying the fuel of wastewater sludge rather than incineration. If possible, wastewater treated with FWDs is introduced into the AD through a separate pipe and tank; the amount of loss can be reduced and the production of CH$_4$ gas can be increased.

### 4. Conclusions

FWDs are practical and relatively inexpensive, and their use can effectively reduce the need for separate bins for collection and disposal and provide a more convenient way to dispose of FW. However, the use of these devices is associated with potentially negative impacts that make their sustainability unclear.

This study analyzed the characteristics of wastewater from FWDs, the effect of pollutants on WWTPs, and the effect of GHG generation, and the results are as follows.

As a result of analyzing the characteristics of wastewater from FWD, it was found that the concentration of pollutants was much higher than that of wastewater. Therefore, in the case of using FWDs, it is a cause that it can impose a burden on separate management and WWTPs.

As a result of analyzing the effect of the wastewater system in Korea due to the introduction of FWDs, it was found that the inflow BOD increased by 51.2% based on the total amount of WWTP brought in and that 86% of WWTP exceeded the appropriate operation rate of 80%. It seems that it is difficult to cope with the introduction of FWDs in the current WWTPs.

As a result of analyzing GHG emissions by FWD, the net GHG was found to increase by 58%. In addition, the GHG emission generated by processing 1 ton of FW was found to be 206.1 tCO$_2$eq. Even if methane is recovered by biogas from WWTP using FWD, the emission of GHG is increased, but this does not mean unreasonable FW disposal using FWDs. To reduce GHG emissions from WWTPs, wastewater treated with FWDs is directly injected into the AD to minimize organic matter loss and reduce energy consumption. If high energy efficiency facilities are actively introduced and GHGs such as sludge dry fuel conversion and composting are reduced, the method using FWDs can be considered as a sustainable treatment method for FW.

**Author Contributions:** Writing—original draft preparation, D.K.; writing—review and editing, C.P. All authors have read and agreed to the published version of the manuscript.

**Funding:** This work was supported by Korea Environment Industry&Technology Institute (KEITI) through "Climate Change R&D Project for New Climate Regime.", funded by Korea Ministry of Environment (MOE) (2022003570003).

**Data Availability Statement:** Related data is based on sewerage statistics but was provided by K-eco (Korea Environment Corporation).

**Conflicts of Interest:** The authors declare no conflict of interest.

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
