# Peer review of "Analysis of the Effect of the Use of Food Waste Disposers on Wastewater Treatment Plant and Greenhouse Gas Emission Characteristics"

_water, doi:10.3390/w15050940_

Round 1
Reviewer 1 Report
I revised the manuscript “Analysis of Effect on Sewerage treatment plant and Green-house Gas Emission Characteristics of Use of Food Waste Disposer” submitted to the Water.
The authors presented the topic on sewerage treatment plant and Green-house Gas Emission Characteristics of Use of Food Waste Disposer.
The interactions of effect of introducing food waste on the sewerage system on increase in inflow load on WWTP with a capacity of 500 ㎥/day is discussed.
Furthermore, authors currented approaches to Greenhouse Gas Emission are introduced. The topic of the article is up to date, the introduction is easy detailed. This problem is relevant for journal scope.
The concept and aim are clearly defined. The presentation and discussion of the presented topicis clear and very detailed.
Suggests supplementing the "Introduction" with information bringing a new scientific contribution. Please provide a review of the literature in this area.The text can be supplemented with information on the use of organic waste in fermentation processes in sewage treatment plants.
To raise the level of paper, please use the articles on the efficiency of fermentation in wastewater treatment plants:
doi: 10.3390/en13226056
doi: 10.3390/en14113157
doi: 10.12912/23920629/122657
doi: 10.3390/fuels2020009
doi: 10.1016/j.jclepro.2021.129107
Please cite more papers from MDPI journals at the last 2-3 years in the similar topic of this research.
Other weaknesses to be corrected:
1. Keywords should be in alphabetical order.
2. I propose to insert a dictionary for abbreviations. No explanation of some abbreviations, e.g. GHG
3. I suggest using Wastewater Treatment Plant (WWTP) instead of Sewerage treatment plant
4. The conclusions should be shorter and more specific. I propose to shorten the conclusions and keep the most important ones. It is best to list 3-4 critical conclusions.
The manuscript follows the formal regulations of MDPI journals.
I suggest the acceptance after minor revision
Author Response
|
To raise the level of paper, please use the articles on the efficiency of fermentation in wastewater treatment plants: doi: 10.3390/en13226056 doi: 10.3390/en14113157 doi: 10.12912/23920629/122657 doi: 10.3390/fuels2020009 doi: 10.1016/j.jclepro.2021.129107 |
The attached thesis and fermentation in various wastewater treatment facilities have been added to the references.
|
|
Please cite more papers from MDPI journals at the last 2-3 years in the similar topic of this research. |
|
|
Other weaknesses to be corrected: 1. Keywords should be in alphabetical order. 2. I propose to insert a dictionary for abbreviations. No explanation of some abbreviations, e.g. GHG 3. I suggest using Wastewater Treatment Plant (WWTP) instead of Sewerage treatment plant 4. The conclusions should be shorter and more specific. I propose to shorten the conclusions and keep the most important ones. It is best to list 3-4 critical conclusions. |
1, 2. Keywords are in alphabetical order, and abbreviations were used as full names at first.
3. WWTP was used instead of STP.
4. The conclusion is relatively simple and clearly revised.
|
Reviewer 2 Report
The subject matter of the manuscript submitted for review is important from both a scientific and practical point of view. Unfortunately, the quality of the article does not allow it to be published in its current state. The methodology of the conducted research is not clearly described, and the assumptions adopted are not reflected in the subsequent parts of the article. References are poor, even though the subject of the work is widely discussed in numerous studies. Also, the way References are prepared is not correct. The Authors have probably done valuable research, so I suggest that they thoroughly revise the manuscript and submit it for re-review. Detailed notes in the attachment.

Author Response
|
Why are some words capitalized and others lowercase? Why Sewerage and not sewage? |
Uppercase and lowercase letters have been unified. Modified to use wastewater as suggested by another reviewer instead of sewage. |
|
Sentence 39-41: What negative research results do the authors write about? Relevant literature references should be listed. |
The negative literature I researched starts at line 55.[18-21] |
|
What the authors mean by "food waste disposal" should be clearly explained, based on the current literature. Figure 1 contains many elements of the process chain, but in the research they are treated separately. It should be emphasized that these are tests for facilities, sewage and food waste typical of Korea, as they are very specific and such results may be misleading for other areas of the world, e.g. Europe. Sentence 76-77: Why was the average of 3 test results determined for BOD, while T-N was taken from one? Line 110: The article is talking about domestic sewage, not domestic STP. |
Lines 42-43 clearly describe the meaning of FWD
|
|
Sentence 195-196: The reference to the LCA methodology is very general and should be expanded or omitted. Table 3: For what inputs are the calculations performed? For those 679 places mentioned in Material and Methods? This should be clearly explained. |
Expressions related to LCA were excluded.
|
|
The “Discussion” chapter is not intended as a substantive discussion with the results of other authors. It is rather an extension of the discussion of the results of own research. This part needs to be thoroughly improved. Why in this part the Authors raise the problem of the presence of impurities, which were not mentioned in the final part (n-Haxen etc.)? |
The findings of the study were discussed and discussed in Results. Changed the discussion to a conclusion.
|
|
Titles of tables and figures should be corrected. They should not start with "This figure shows..." etc. |
|
|
The subject matter discussed in the manuscript is up-to-date and numerous studies are conducted in this field. Therefore, I propose to expand the literature review, to discuss with the results of other researchers, and thus - to expand the References. All literature items should be spelled correctly. Journal titles should be written in capital letters, using their abbreviations. You must provide the DOI of articles that have it. Ref. 15 – authors? |
A lot of literature has been added. The reference format has been modified. |
|
Abbreviations should be explained where they first appear in the text |
It's corrected. |
|
Food – food (lowercase letter) |
It's corrected. |
|
What is RAS, WAS – explain! |
It's corrected. |
|
In this photo you can see mainly cards with markings .. |
Could this be a problem? It was edited because it was marked in Korean. |
|
Names of waste components – in lowercase letters. Spaces before and after |
It's corrected. |
|
What is STP A? |
Corrected by typo. |
|
What is 1) ? |
Corrected by typo. |
|
Start new sentences with a capital letter |
It's corrected. |
|
Spaces should be placed before parenthese |
It's corrected. |
|
After [15] should be dot, not comma |
It's corrected. |
|
Dot after Fig. |
It's corrected. |
|
Add PCCC guidelines to References |
References are attached. |
Reviewer 3 Report
Review report on the paper titled “Analysis of Effect on Sewerage treatment plant and Greenhouse Gas Emission Characteristics of Use of Food Waste Disposer”
Overall, the quality of the manuscript is very low and lacks presentation of any innovation or interesting finding. There are serious flaws in methodology. The English language needs extensive editing. Specific comments are made in the appended manuscript, so that the authors can further improve the manuscript.

Author Response
|
Need to re-write the introduction addressing the topic in detail. Standard Introduction/literature review length is 2500 to 3000 words. |
Edited by adding references. |
|
please first introduce what is FWD and its process in 3-5 lines. |
Lines 42-43 clearly describe the meaning of FWD |
|
How it is a sustainable method? discuss with ref. |
As a reference, we have described why it is not sustainable. |
|
Please establish the background of the research, and novelty paragraph by identifying the research gap |
As for the background of the study, it was originally suggested that this study was conducted to examine the effect of FWD, since the greenhouse gas review considering the food characteristics of Korea had not been properly conducted.
|
|
need the details of experimental procedure. Not sure, what is the experiment on? |
In the experiment, water was poured into 500 g of food, and the dehydrated liquid or the wastewater treated with fwd was analyzed. The analysis procedure was described in more detail.
|
|
Need details, as this is not common in many countries. |
Food waste disposer and grinder are used a lot, but in general, the above terms seem to be used more. Please let me know if there is. |
|
What are these equipment? mention in detail. |
The device is a product, which is not marked as it is a sensitive matter in Korea. |
|
Need ref(“the national waste statistical survey 5th). |
References are attached. |
|
what is this?(so it may be necessary to expand) |
An increase in pollutants means that it is necessary to increase the capacity of the sewage treatment plant. |
|
what is this?(679 places of 500 71 ㎥/day or more).) |
This means that there are 679 facilities with a processing capacity of 500M3 or more.
|
|
show detailed calculation in Appendix |
References are attached. |
|
Ref (IPCC guidelines(2006)) |
References are attached. |
|
need detailed of the process of calculation, including equations |
I have attached the formula. |
|
How it is related to GHG emission...show in the figure |
It is stated that greenhouse gases are released from water treatment and biogas in the figure. |
|
is this analysis data based on the experiment conducted? |
no. Operational data and FW generated in domestic households were used.
|
|
how calculated? show this along with other calculation(8,718,673 KW/day) |
The data is operational data and is the daily power consumption of 679 facilities. |

Round 2
Reviewer 2 Report
The manuscript was significantly improved, the authors took into account the comments contained in the reviews, which increased the substantive quality of the way of presenting the research results. The article is suitable for publication in its current form.
Author Response
Thanks to your help, the article has been improved a lot. Thank you for your hard work.
Reviewer 3 Report
Unfortunately, the manuscript needs further modification. The authors have addressed some comments, but still some comments need to be addressed.
1. Introduction: While some information is added, still there is no description on Greenhouse Gas emissions from WWTP and how it is related to FW disposer. This should be included in the introduction section.
2. Methodology: Still it is not clear how the experimental part (Characterization of three types of FWD generated wastewater are linked to GHG emission calculation.
3. Results: Table 3: The authors should clarify the source of the "amount and characteristics of the influent, effluent and sludge" used in the table. Please show the data table and sample calculation for the scientific replication of your claim.
Author Response
Unfortunately, the manuscript needs further modification. The authors have addressed some comments, but still some comments need to be addressed.
- Introduction: While some information is added, still there is no description on Greenhouse Gas emissions from WWTP and how it is related to FW disposer. This should be included in the introduction section.
- It was described that food increases the efficiency of anaerobic digestion by increasing the concentration of organic matter in wastewater with low organic matter.(line 52~)
- Methodology: Still it is not clear how the experimental part (Characterization of three types of FWD generated wastewater are linked to GHG emission calculation.
- Greenhouse gas emissions were not calculated through the experimental part. The experiment was conducted to confirm that the concentration of contaminants in wastewater from fwd is higher than that of normal wastewater.
- For greenhouse gas emissions, sewage treatment volume, sewage treatment population, and BOD inflow concentration were used in Korea's sewerage statistics. Food waste is the amount of food waste generated per population in each region.
- Results: Table 3: The authors should clarify the source of the "amount and characteristics of the influent, effluent and sludge" used in the table. Please show the data table and sample calculation for the scientific replication of your claim.
- I have attached the statistics used and an example of calculating the amount of food waste input.
Round 3
Reviewer 3 Report
Improved.